# Beta Equilibrium under Neutron Star Merger Conditions

**Mark G. Alford** [1], **Alexander Haber** [1,*], **Steven P. Harris** [2] **and Ziyuan Zhang** [1,3]

1 Physics Department, Washington University in Saint Louis, Saint Louis, MO 63130, USA; alford@wustl.edu (M.G.A.); ziyuan.z@wustl.edu (Z.Z.)
2 Institute for Nuclear Theory, University of Washington, Seattle, WA 98195, USA; harrissp@uw.edu
3 McDonnell Center for the Space Sciences, Washington University in St. Louis, St. Louis, MO 63130, USA
* Correspondence: ahaber@physics.wustl.edu

**Abstract:** We calculate the nonzero-temperature correction to the beta equilibrium condition in nuclear matter under neutron star merger conditions, in the temperature range $1\,\text{MeV} < T \lesssim 5\,\text{MeV}$. We improve on previous work using a consistent description of nuclear matter based on the IUF and SFHo relativistic mean field models. This includes using relativistic dispersion relations for the nucleons, which we show is essential in these models. We find that the nonzero-temperature correction can be of order 10 to 20 MeV, and plays an important role in the correct calculation of Urca rates, which can be wrong by factors of 10 or more if it is neglected.

**Keywords:** nuclear matter; neutron star merger; beta equilibration; weak interaction

## 1. Introduction

Nuclear matter in neutron stars settles into beta equilibrium, meaning that the proton fraction is in equilibrium with respect to the weak interactions. In this paper, we will study the conditions for beta equilibrium in ordinary nuclear matter (where all the baryon number is contributed by neutrons ($n$) and protons ($p$)) in the temperature range $1\,\text{MeV} \lesssim T \lesssim 5\,\text{MeV}$. This regime, which arises in neutron star mergers [1–4], is cool enough so that neutrinos are not trapped, but warm enough so that there are corrections to the low-temperature equilibrium condition. It has previously been shown [5] that in this regime the full beta equilibrium condition is

$$\mu_n = \mu_p + \mu_e + \Delta\mu \,, \tag{1}$$

where $\Delta\mu$ is a correction that arises from the violation of detailed balance (neutrino transparency) and the breakdown of the Fermi surface approximation (see Section 2). In nuclear matter in the temperature regime discussed here, the proton fraction will equilibrate towards the value given by Equation (1). Even if equilibrium is not reached on the timescale of a merger, one needs to know the correct equilibration condition in order to analyze phenomena associated with this relaxation process, such as bulk viscosity and neutrino emission. At low temperatures ($T \ll 1\,\text{MeV}$) $\Delta\mu$ is negligible, but in the temperature regime under consideration here it has been estimated to be up to tens of MeV [5]. The calculation in Ref. [5] went beyond the Fermi surface approximation by performing the phase space integral for the equilibration rate over the entire momentum space. However, it used a very crude model of the in-medium nucleons, assigning them their vacuum mass and assuming that their kinematics remained nonrelativistic at all densities.

In this paper, we improve on the analysis of Ref. [5]. We treat nuclear matter consistently using relativistic mean field models [6,7] with fully relativistic dispersion relations for the nucleons. We show that this makes a considerable difference to the beta equilibration rates because in these models the nucleons at the Fermi surface become relativistic at densities of a few times nuclear saturation density $n_0$. We calculate the direct Urca rate

using the entire weak-interaction matrix element rather than its nonrelativistic limit, and evaluate the full phase space integral.

Other authors have evaluated direct Urca phase space integrals in calculations of the direct Urca rate, the neutrino emissivity, or the neutrino mean free path. Fully relativistic computations of direct Urca phase space integrals are uncommon in the literature, but they do appear. Refs. [8–11] calculate the neutrino mean free path using a fully relativistic formalism, while integrating over the full phase space. Ref. [10] calculates the direct Urca electron capture rate using a fully relativistic formalism and performs the full phase space integration. Although these calculations perform the full integration over phase space, they focus on high temperatures ($T \gtrsim 5$ MeV) where neutrinos are trapped and where the direct Urca threshold is blurred over a wide density range. In this temperature regime, which can be reached in mergers as well [1,12–14], beta equilibrium is given by

$$\mu_n + \mu_\nu = \mu_p + \mu_e \,, \tag{2}$$

with $\mu_\nu$ being the neutrino chemical potential. As discussed in more detail in Section 2, the neutrino-trapped beta equilibration condition does not require an additional finite-temperature correction. This paper will examine the phase space integral at lower temperatures where the direct Urca threshold is apparent and a key feature in the physics of beta equilibration or neutrino emission.

Other works use the relativistic formalism, but assume the nuclear matter is strongly degenerate (using the Fermi surface approximation, described below), and thus their results have a sharp direct Urca threshold density [15–17]. Ref. [18] uses the Fermi surface approximation, but develops a way to incorporate the finite 3-momentum of the neutrino, slightly blurring the threshold at finite temperature. Some works do the full phase space integration, but use nonrelativistic approximations for the matrix element and nucleon dispersion relations [5,19–21]. The vast majority of calculations use nonrelativistic approximations of the matrix element and the nucleon dispersion relations, together with the Fermi surface approximation [22–34]. All of these calculations are approximations of the full phase space integration using the fully relativistic formalism. Under certain conditions, the approximations match well with the full calculation, and have the advantage of being simple.

In Section 3 we introduce the two relativistic mean field models, IUF and SFHo, that we use. Section 4 describes our calculation of the rate of direct Urca processes, where we integrate over the entire phase space in order to include contributions from the region that would be kinematically forbidden in the low-temperature limit. Section 5 describes our calculation of the modified Urca contribution to the rate, where we use the Fermi surface approximation since there is no kinematically forbidden region for those processes in the density range that we consider. Section 6 presents our results, and Section 7 provides our conclusions.

We work in natural units, where $\hbar = c = k_B = 1$.

## 2. Beta Equilibration

Beta equilibration in $npe^-$ matter is established by the Urca processes [35]. The modified Urca processes

$$N + n \rightarrow N + p + e^- + \bar{\nu} \tag{3}$$
$$N + p + e^- \rightarrow N + n + \nu,$$

(here, $N$ represents a "spectator" neutron or proton) operate at all densities in the core of the neutron star. In uniform $npe^-$ matter, the proton-spectator modified Urca process only operates at densities where $x_p > 1/65$ [25,31], though this condition is only violated (if

ever) in the inner crust of neutron stars [36] where the matter is not uniform and thus the calculations in this paper would not apply. The direct Urca processes

$$n \rightarrow p + e^- + \bar{\nu} \tag{4}$$
$$p + e^- \rightarrow n + \nu,$$

are exponentially suppressed when the temperature is much less than the Fermi energies and the density is in the range where $k_{Fn} > k_{Fp} + k_{Fe}$. In nuclear matter, the proton fraction rises as the density rises above $n_0$ and eventually may reach a "direct Urca threshold" where $k_{Fn} = k_{Fp} + k_{Fe}$. Above this threshold density beta equilibration is dominated by direct Urca, since (when kinematically allowed) it is faster than modified Urca.

In nuclear matter at temperatures greater than, say, 10 MeV, the neutrino mean free path is short and the nuclear matter system (for example, a protoneutron star) is neutrino-trapped and has conserved lepton number $Y_L = (n_e + n_\nu)/n_B$. In this case, the Urca processes (3) and (4) can proceed forward and backward, as the nuclear matter contains a population of neutrinos (or antineutrinos). In beta equilibrum, the forward and reverse processes have equal rates (detailed balance), and the beta equilibrium condition is given by balancing the chemical potentials of the participants in the equilibration reactions [6,37]

$$\mu_n + \mu_\nu = \mu_p + \mu_e \qquad (\nu\text{-trapped}). \tag{5}$$

In cooler nuclear matter, at the temperatures considered in this work, the neutrino mean free path is comparable to or longer than the system size and therefore neutrinos are not in thermodynamic equilibrium: they escape from the star. Neutrinos can then occur in the final state but not the initial state of the Urca processes. Beta equilibrium is still achieved, but now by a balance of the neutron decay and the electron capture processes. However, the principle of detailed balance is not applicable because electron capture is not the time-reverse of neutron decay.

There is then no obvious equilibrium condition that can be written down a priori. In the limit of low temperature ($T \ll 1\,\text{MeV}$) the Fermi surface approximation becomes valid: the particles participating in the Urca processes are close to their Fermi surfaces, and the neutrino carries negligible energy $\sim T$. The beta equilibrium condition can then be obtained by neglecting the neutrino, so that neutron decay and electron capture are just different time orderings of the same process $n \leftrightarrow p\,e^-$, and detailed balance gives

$$\mu_n = \mu_p + \mu_e \qquad (\text{low temperature, } \nu\text{-transparent}). \tag{6}$$

The same condition on the chemical potentials can be reached by examining the phase space integrals for the direct Urca neutron decay and electron capture rates, taking the limit where the neutrino energy and momentum go to zero [38]. At temperatures $T \gtrsim 1\,\text{MeV}$ corrections to the Fermi surface approximation start to become significant, particularly for the protons whose Fermi energy is in the 10 MeV range. Then one cannot neglect the finite-temperature correction to (6)

$$\mu_n = \mu_p + \mu_e + \Delta\mu \quad (\text{general, } \nu\text{-transparent}). \tag{7}$$

The correction $\Delta\mu$ is a function of density and temperature, and its value in beta equilibrium is found by explicitly calculating the neutron decay and electron capture rates and adjusting $\Delta\mu$ so that they balance [5] (see also [39], where a similar calculation was done in the context of a hot plasma). In this paper, we perform that calculation.

For weak interactions we use the Fermi effective theory, which is an excellent approximation at nuclear energy scales. The main approximations arise in our treatment of the strong interaction. To describe nuclear matter and the nucleon excitations we use two different relativistic mean field models, both consistent with known phenomenology and chosen to illustrate a plausible range of behaviors. We describe these models in Section 3.

For the modified Urca process we model the nucleon-nucleon interaction with one-pion exchange [31,40].

## 3. Nuclear Matter Models

We will use two different equations of state, IUF [41] and SFHo [42], to calculate the Urca rates and the nonzero-temperature correction $\Delta\mu$. These are both consistent at the $2\sigma$ level with observational constraints on the maximum mass and the radius of neutron stars.

IUF predicts a maximum mass of neutron star to be $1.95 M_\odot$, and SFHo predicts $2.06 M_\odot$. Both are consistent with the observed limits, which are:

- $M_{\mathrm{max}} > 2.072^{+0.067}_{-0.066}\, M_\odot$ from NICER and XMM analysis of PSR J0740+6620 [43];
- $M_{\mathrm{max}} = 1.928^{+0.017}_{-0.017}\, M_\odot$ from NANOGrav analysis of PSR J1614-2230 [44];
- $M_{\mathrm{max}} = 2.01^{+0.14}_{-0.14}\, M_\odot$ from pulsar timing analysis of PSR J0348+0432 [45].

For the radius of a star of mass $2.06\, M_\odot$, SFHo predicts $R = 10.3\,\mathrm{km}$, consistent with $R = 12.39^{+1.30}_{-0.98}\,\mathrm{km}$ from NICER and XMM analysis of PSR J0740+6620 [43]. For the radius of a $1.4\, M_\odot$ neutron star, IUF predicts $R = 12.7\,\mathrm{km}$ and SFHo predicts $R = 11.9\,\mathrm{km}$, consistent with $R = 11.94^{+0.76}_{-0.87}\,\mathrm{km}$ obtained by a combined analysis of X-ray and gravitational wave measurements of PSR J0740+6620 in Ref. [46].

It is still not determined whether there is a direct Urca threshold or not in nuclear matter at neutron star densities [47–51], so we choose one equation of state (IUF) with a threshold at $4.1 n_0$ and one (SFHo) with no threshold, as shown in Figure 1. Our approach could be applied to any equation of state where the beta process rates can be calculated. As we will see in Section 6.3, the density dependence of the momentum surplus $k_{Fp} + k_{Fe} - k_{Fn}$ is an important factor in the behavior of the direct Urca rates at low temperature, but the density dependence of the nucleon effective masses and Fermi momenta has a noticeable impact as well.

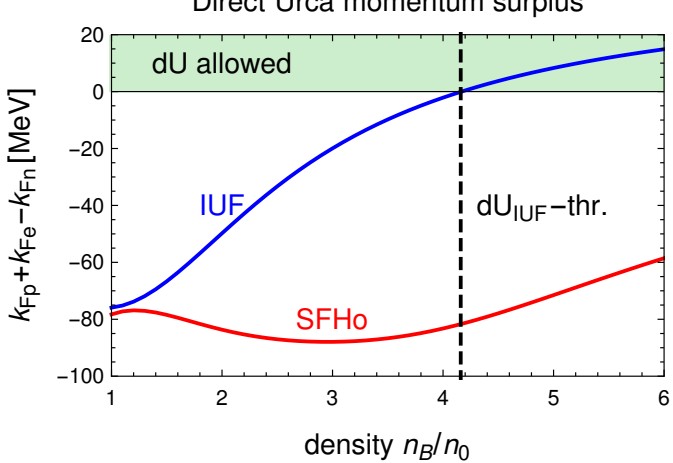

**Figure 1.** Direct Urca momentum surplus $k_{Fp} + k_{Fe} - k_{Fn}$ for IUF and SFHo equations of state at $T = 0$. When the surplus is negative, direct Urca is forbidden. IUF has an upper density threshold above which direct Urca is allowed; SFHo does not.

The coupling constants for SFHo are shown in Appendix A. Notice that the constants are taken from the online CompOSE database (https://compose.obspm.fr/, accessed on 27 April 2021), and are different from the values provided in Ref. [42].

A key feature of our calculation is that we use the full relativistic dispersion relations for the nucleons. In Figures 2 and 3 we illustrate the importance of this in relativistic mean field theories, where the nucleon effective mass drops rapidly with density. Although the precipitous drop in the nucleon Dirac effective mass with increasing density is a common feature in relativistic mean field theories [52,53], we note that in two recent treatments that go beyond the mean field approximation, the drop in the effective mass was not as

dramatic [54,55]. We plot the Dirac effective mass [56] and the Fermi momentum of the neutrons and protons in these two EoSs. Although around nuclear saturation density $n_0$, the nucleons are nonrelativistic, as the density rises to several times $n_0$, the nucleon effective mass has dropped significantly below its vacuum value. Neutrons on their Fermi surface become relativistic at $2 - 3n_0$, while protons on their Fermi surface remain nonrelativistic until the density rises to $3 - 6n_0$. In Figures 4 and 5, we show that using a nonrelativistic approximation would lead to Urca rates that are incorrect by about an order of magnitude, although for direct Urca neutron decay the discrepancy can be many orders of magnitude.

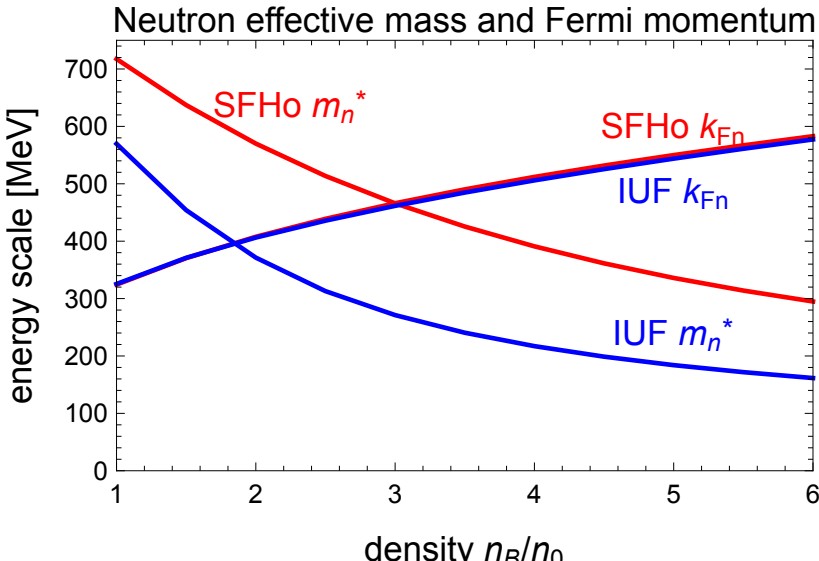

**Figure 2.** Density dependence of the neutron's (Dirac) effective mass and Fermi momentum for the IUF and SFHo EoSs, showing that neutrons at the Fermi surface become relativistic at densities above 2 to 3 $n_0$.

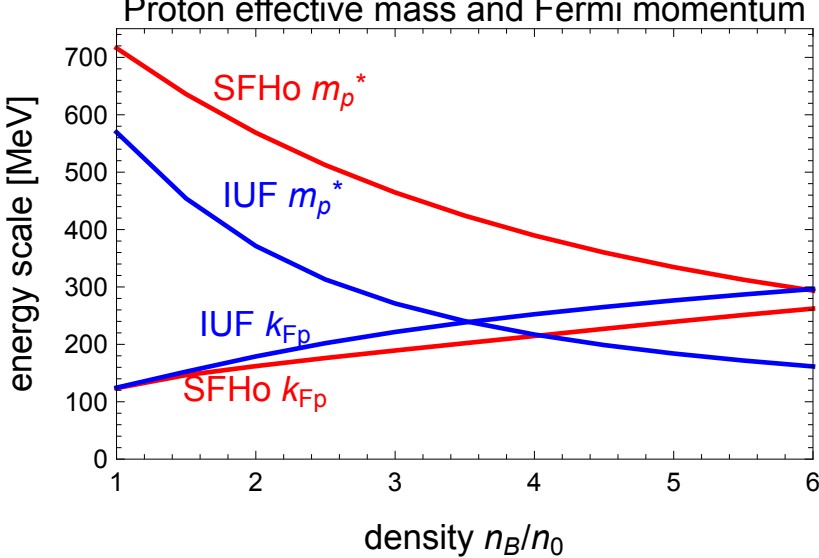

**Figure 3.** Density dependence of the proton's (Dirac) effective mass and Fermi momentum for the IUF and SFHo EoSs, showing that protons at the Fermi surface become relativistic starting at densities between $3 - 6n_0$.

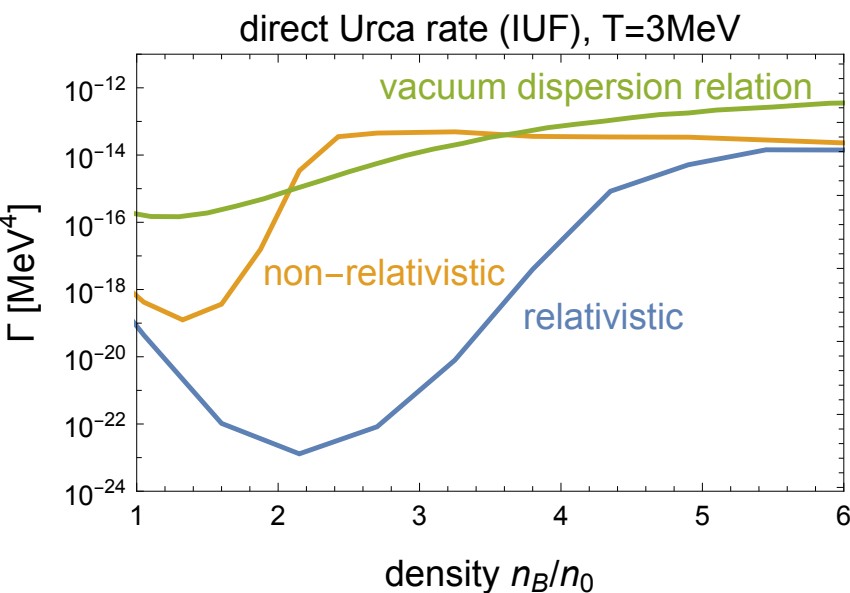

**Figure 4.** Direct Urca neutron decay rate calculated using relativistic, nonrelativistic and the vacuum dispersion relations at $T = 3\,\text{MeV}$ for IUF.

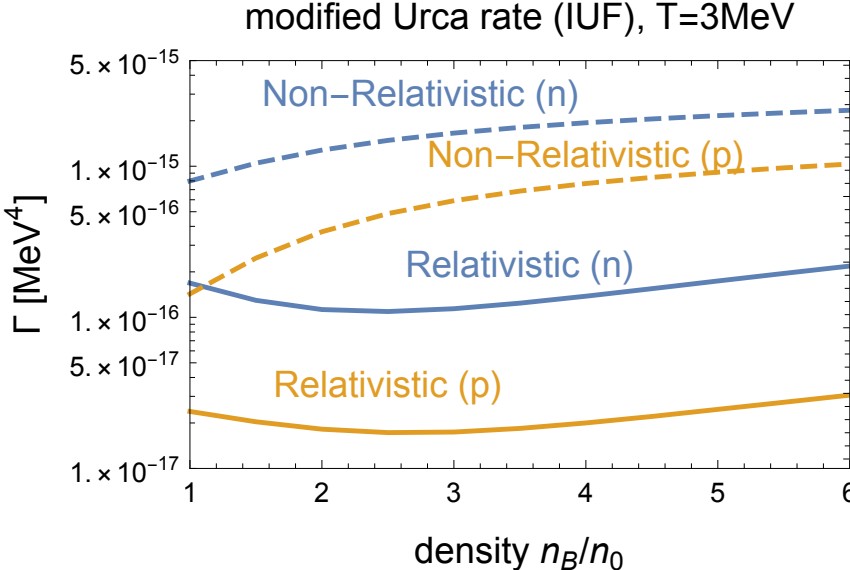

**Figure 5.** Modified Urca rate calculated using relativistic and nonrelativistic dispersion relations at $T = 3\,\text{MeV}$ for IUF. (n) stands for neutron-spectator modified Urca and (p) stands for proton-spectator modified Urca.

## 4. Beta Equilibration via Direct Urca

We calculate the in-medium direct Urca rates for neutron decay and electron capture using the relativistic weak-interaction matrix element and the relativistic dispersion relations for the nucleons and electrons. We also integrate over the full momentum phase space, not relying on the Fermi surface approximation. This is important because in the "dUrca-forbidden" density range the Fermi surface approximation would say the direct Urca rate is zero, so nonzero-temperature corrections are the leading contribution. These become significant (comparable to modified Urca) at the temperatures of interest here, $T \gtrsim 1\,\text{MeV}$ [5].

In relativistic mean field models the dispersion relations for the neutrons, protons, and electrons are

$$E_n = \underbrace{\sqrt{m_n^{*2} + k_n^2}}_{E_n^*} + U_n$$

$$E_p = \underbrace{\sqrt{m_p^{*2} + k_p^2}}_{E_p^*} + U_p \qquad (8)$$

$$E_e = \sqrt{m_e^2 + k_e^2}$$

$$E_\nu = k_\nu,$$

where the nucleons' effective mass $m_i^*$ and energy shift $U_i$ depend on density and temperature [10]. The unshifted energies $E_i^*$ arise in the phase space normalization and the Dirac traces [9].

### 4.1. Neutron Decay

The direct Urca neutron decay rate is [31,57]

$$\Gamma_{nd} = \int \frac{d^3k_n}{(2\pi)^3} \frac{d^3k_p}{(2\pi)^3} \frac{d^3k_e}{(2\pi)^3} \frac{d^3k_\nu}{(2\pi)^3} f_n (1 - f_p)(1 - f_e) \frac{\sum |M|^2}{(2E_n^*)(2E_p^*)(2E_e)(2E_\nu)}$$
$$(2\pi)^4 \delta^{(4)}(k_n - k_p - k_e - k_\nu). \qquad (9)$$

For a more detailed explanation of this expression and its evaluation, see Appendix B. As described there, it can be reduced to 5-dimensional momentum integral (43)

$$\Gamma_{nd} = \frac{G^2}{16\pi^6} \int_0^\infty dk_n \int_0^{k_p^{max}} dk_p \int_0^{k_e^{max}} dk_e k_n^2 k_p^2 k_e^2 f_n (1 - f_p)(1 - f_e) \Theta(E_\nu)$$
$$\int_{z_p^{min}}^{z_p^{max}} dz_p \int_{z_e^-}^{z_e^+} dz_e \frac{4E_\nu \mathcal{M}_{\phi_0}}{\sqrt{S^2 - (E_\nu^2 - R)^2}}, \qquad (10)$$

where $R$, $S$, and $\mathcal{M}_{\phi_0}$ are defined in Equations (24)–(26). The antineutrino energy $E_\nu$ is given by

$$E_\nu = E_n - E_p - E_e, \qquad (11)$$

which becomes a function of the remaining integration variables, $k_n$, $k_p$, and $k_e$. Please note that there are Fermi-Dirac distributions for the neutrons, proton vacancies, and electron vacancies, but none for the neutrinos because we work in the neutrino-transparent regime where neutrinos escape from the star and do not form a Fermi gas. We evaluate this integral numerically using a Monte-Carlo algorithm.

### 4.2. Electron Capture

The expression for the electron capture rate can be obtained from that for neutron decay (A10) by making the following changes: (1) the energy-momentum delta function now corresponds to the process $p\, e^- \to n\, \nu$, and (2) there are Fermi-Dirac distributions for proton and electron particles, and neutron vacancies,

$$\Gamma_{ec} = \int \frac{d^3k_n}{(2\pi)^3} \frac{d^3k_p}{(2\pi)^3} \frac{d^3k_e}{(2\pi)^3} \frac{d^3k_\nu}{(2\pi)^3} (1 - f_n) f_p f_e \frac{\sum |M|^2}{(2E_n^*)(2E_p^*)(2E_e)(2E_\nu)}$$
$$(2\pi)^4 \delta^{(4)}(k_p + k_e - k_n - k_\nu). \qquad (12)$$

Evaluating this expression takes us through the same steps as for neutron decay, except that the neutrino energy is now

$$E_\nu = E_p + E_e - E_n,$$ (13)

and the requirement that this be positive leads to different limits on the momentum integrals,

$$\Gamma_{\rm ec} = \frac{G^2}{16\pi^6} \int_0^\infty dk_n \int_0^\infty dk_p \int_0^\infty dk_e k_n^2 k_p^2 k_e^2 f_n (1 - f_p)(1 - f_e) \Theta(E_\nu)$$
$$\int_{z_p^{\rm min}}^{z_p^{\rm max}} dz_p \int_{z_e^-}^{z_e^+} dz_e \frac{4 E_\nu \mathcal{M}_{\phi_0}}{\sqrt{S^2 - (E_\nu^2 - R)^2}}.$$ (14)

## 5. Beta Equilibration via Modified Urca

We calculate the rate of the modified Urca processes (3) using the relativistic dispersion relations of the nucleons in the phase space integration, but unlike the direct Urca rate we do not perform the phase space integration exactly, which would be difficult because the involvement of the spectator particles would lead to an 11-dimensional numerical integral over momentum. Instead we use the Fermi surface approximation. This is reasonable for modified Urca as long as the Fermi surfaces are not too thermally blurred, i.e. when the temperature is below the lowest Fermi kinetic energy, which is that of the proton. The modified Urca processes do not have a density threshold in the range of densities we consider here (see Section 2), so the Fermi surface approximation never predicts a vanishing rate. In this work we explore the temperature range $1\,{\rm MeV} < T < 5\,{\rm MeV}$, and the proton's Fermi kinetic energy is at least $10\,{\rm MeV}$ in the density range $n > n_0$, so the Fermi surface approximation is justified for modified Urca rates. The first paragraph of Section 4 contains a discussion of why we need to go beyond the Fermi surface approximation in our direct Urca rate calculations. For the matrix elements that arise in modified Urca (44) and (59), we use the standard results (see, e.g., [31]), which were calculated assuming nonrelativistic nucleons. It has been pointed out [58] that the standard calculation of the modified Urca matrix element [40], which we use here, is based on a very crude approximation for the propagator of the internal off-shell nucleon. A more accurate treatment would lead to different modified Urca rates and shift our predicted values of $\Delta\mu$; we defer such a calculation to future work.

### 5.1. Neutron Decay

Modified Urca can proceed with either a neutron spectator or a proton spectator. From Fermi's Golden rule, we have the rate for the neutron decay process

$$\Gamma_{mU,nd} = \int \frac{d^3k_n}{(2\pi)^3} \frac{d^3k_p}{(2\pi)^3} \frac{d^3k_e}{(2\pi)^3} \frac{d^3k_\nu}{(2\pi)^3} \frac{d^3k_{N_1}}{(2\pi)^3} \frac{d^3k_{N_2}}{(2\pi)^3} \left( s \frac{\sum |M|^2}{2^6 E_n^* E_p^* E_e E_\nu E_{N_1}^* E_{N_2}^*} \right)$$
$$(2\pi)^4 \delta^{(4)}(k_n + k_{N_1} - k_p - k_e - k_\nu - k_{N_2}) f_n f_{N_1}(1 - f_p)(1 - f_e)(1 - f_{N_2}).$$ (15)

Here, $s = 1/2$ because of the identical particles appearing in the process. $N_1$ and $N_2$ are neutrons in the n-spectator process and for the p-spectator neutron decay process, $N_1$ and $N_2$ are protons. The matrix element is different for each process see Equations (44) and (59). The detailed derivation of the modified Urca rates is in Appendix C. For n-spectator neutron decay, allowing the system to deviate from the low-temperature beta equilibrium condition (6) by amount

$$\zeta = \frac{\mu_n - \mu_p - \mu_e}{T},$$ (16)

we obtain

$$\Gamma_{mU,nd(n)}(\xi) = \frac{7}{64\pi^9}G^2 g_A^2 f^4 \frac{(E_{Fn}^*)^3 E_{Fp}^*}{m_\pi^4} \frac{k_{Fn}^4 k_{Fp}}{(k_{Fn}^2 + m_\pi^2)^2} F(\xi) T^7 \theta_n, \tag{17}$$

where $f \approx 1$ is the N-$\pi$ coupling [31],

$$\begin{aligned}
F(\xi) \equiv &- (\xi^4 + 10\pi^2\xi^2 + 9\pi^4)\mathrm{Li}_3(-e^\xi) + 12(\xi^3 + 5\pi^2\xi)\mathrm{Li}_4(-e^\xi) \\
&- 24(3\xi^2 + 5\pi^2)\mathrm{Li}_5(-e^\xi) + 240\xi\mathrm{Li}_6(-e^\xi) - 360\mathrm{Li}_7(-e^\xi),
\end{aligned} \tag{18}$$

and

$$\theta_n \equiv \begin{cases} 1 & k_{Fn} > k_{Fp} + k_{Fe} \\ 1 - \dfrac{3}{8}\dfrac{(k_{Fp} + k_{Fe} - k_{Fn})^2}{k_{Fp}k_{Fe}} & k_{Fn} < k_{Fp} + k_{Fe}. \end{cases} \tag{19}$$

The functions $\mathrm{Li}_n(x)$ are polylogarithms of order $n$ [59]. For p-spectator neutron decay, we obtain

$$\Gamma_{mU,nd(p)}(\xi) = \frac{1}{64\pi^9}G^2 g_A^2 f^4 \frac{(E_{Fp}^*)^3 E_{Fn}^*}{m_\pi^4} \frac{(k_{Fn} - k_{Fp})^4 k_{Fn}}{((k_{Fn} - k_{Fp})^2 + m_\pi^2)^2} F(\xi) T^7 \theta_p, \tag{20}$$

where

$$\theta_p \equiv \begin{cases} 0 & \text{if } k_{Fn} > 3k_{Fp} + k_{Fe} \\[2mm] \dfrac{(3k_{Fp} + k_{Fe} - k_{Fn})^2}{k_{Fn}k_{Fe}} & \text{if } \begin{array}{l} k_{Fn} > 3k_{Fp} - k_{Fe} \\ k_{Fn} < 3k_{Fp} + k_{Fe} \end{array} \\[3mm] 4\dfrac{3k_{Fp} - k_{Fn}}{k_{Fn}} & \text{if } \begin{array}{l} 3k_{Fp} - k_{Fe} > k_{Fn} \\ k_{Fn} > k_{Fp} + k_{Fe} \end{array} \\[3mm] \left(2 + 3\dfrac{2k_{Fp} - k_{Fn}}{k_{Fe}} - 3\dfrac{(k_{Fp} - k_{Fe})^2}{k_{Fn}k_{Fe}}\right) & \text{if } k_{Fn} < k_{Fp} + k_{Fe}. \end{cases} \tag{21}$$

### 5.2. Electron Capture

The electron capture modified Urca rate can be obtained in a similar way to neutron decay, by changing the sign of the neutrino 4-momentum in the energy-momentum delta function and interchanging the particle and hole Fermi-Dirac factors,

$$\begin{aligned}
\Gamma_{mU,ec} = \int &\frac{d^3k_n}{(2\pi)^3} \frac{d^3k_p}{(2\pi)^3} \frac{d^3k_e}{(2\pi)^3} \frac{d^3k_\nu}{(2\pi)^3} \frac{d^3k_{N_1}}{(2\pi)^3} \frac{d^3k_{N_2}}{(2\pi)^3} \left(s\frac{\sum|M|^2}{2^6 E_n^* E_p^* E_e E_\nu E_{N_1}^* E_{N_2}^*}\right) \\
&(2\pi)^4\delta^{(4)}(k_p + k_e + k_{N_1} - k_n - k_\nu - k_{N_2})f_p f_e f_{N_1}(1 - f_n)(1 - f_{N_2}).
\end{aligned} \tag{22}$$

Through a similar calculation, we find that the modified Urca neutron decay and electron capture rates in the Fermi surface approximation are related by

$$\Gamma_{mU,ec(n)}(\xi) = \Gamma_{mU,nd(n)}(-\xi), \tag{23}$$

and

$$\Gamma_{mU,ec(p)}(\xi) = \Gamma_{mU,nd(p)}(-\xi). \tag{24}$$

## 6. Results

### 6.1. Beta Equilibrium at Nonzero Temperature

Figures 6 and 7 show our final results for the nonzero-temperature correction $\Delta\mu$ required to achieve beta equilibrium, for the IUF and SFHo equations of state, respectively. The key features are

- At low temperatures $T \lesssim 1$ MeV, the Fermi surface approximation is valid and beta equilibrium is achieved with a negligible correction $\Delta\mu$ (see Section 2).
- At the temperature rises through the neutrino-transparent regime, the value of $\Delta\mu$ rises.
- We only provide results for temperatures up to 5 MeV because at temperatures of around 5 to 10 MeV the neutrino mean free path will become smaller than the star, invalidating our assumption of neutrino transparency.
- The figures indicate that the nonzero-temperature correction reaches values of 10 to 20 MeV before neutrino trapping sets in.
- The density dependence of $\Delta\mu$ appears very different for different EoSs. For IUF the largest values are reached at moderate densities, near the direct Urca threshold. For SFHo, $\Delta\mu$ has a minimum at those densities.

In the rest of this section we will explain these features of our results.

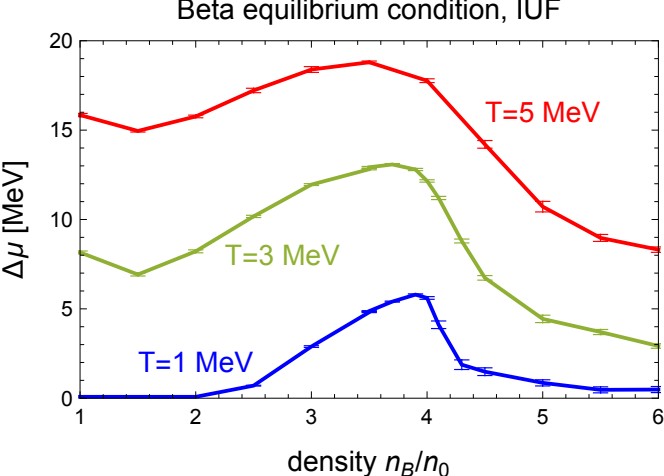

**Figure 6.** Nonzero-temperature correction $\Delta\mu$ required for beta equilibrium Equation (7) with the IUF EoS.

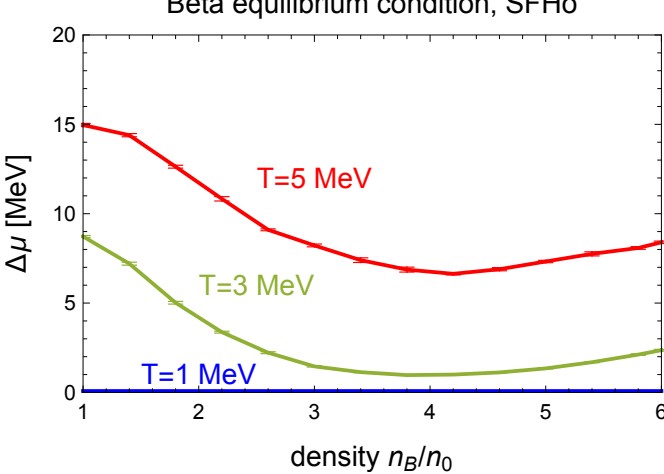

**Figure 7.** Nonzero-temperature correction $\Delta\mu$ required for beta equilibrium Equation (7) with the SFHo EoS.

The temperature dependence follows from the breakdown of the Fermi surface approximation. At $T \lesssim 1\,\mathrm{MeV}$ the Urca processes are dominated by modes close to the Fermi surfaces of the neutron, protons, and electrons. The energy of the emitted neutrino is of order $T$ which is negligible, so the direct Urca process is effectively $n \leftrightarrow p\,e^-$, for which the equilibrium condition is $\mu_n = \mu_p + \mu_e$, i.e., $\Delta\mu = 0$. As the temperature approaches the Fermi energy of the protons, the Fermi surface approximation breaks down. Modes far from the proton and electron Fermi surfaces begin to play a role, and the energy of the emitted neutrino becomes important. The processes that establish beta equilibrium, $n \to p\,e^-\,\bar{\nu}_e$ and $p\,e^- \to n\,\nu_e$, are not related by time reversal, so the principle of detailed balance does not apply. This means that even below the direct Urca threshold density, direct Urca processes can be fast enough and sufficiently different in their rates to require a correction $\Delta\mu$ to bring them into balance. As we will explain below, at $\Delta\mu = 0$ electron capture is much less suppressed than neutron decay, requiring a positive value of $\Delta\mu$ to decrease the proton fraction and equalize the rates.

The density dependence of the correction $\Delta\mu$ is more complicated, depending on specific features of the equations of state. We will discuss this in more detail below.

*6.2. Urca Rates*

Figure 8 illustrates how, without a nonzero-temperature correction $\Delta\mu$ (dashed lines), the neutron decay (nd) and electron capture (ec) rates become very different when the temperature rises to $3\,\mathrm{MeV}$. For both EoSs, electron capture is significantly faster than neutron decay, so a positive $\Delta\mu$ will be required to balance the rates and establish beta equilibrium (solid lines). This is because a positive $\Delta\mu$ reduces the proton fraction. The resultant change in the phase space near the neutron and proton Fermi surfaces enhances the neutron decay rate and suppresses electron capture, bringing the two processes into balance with each other.

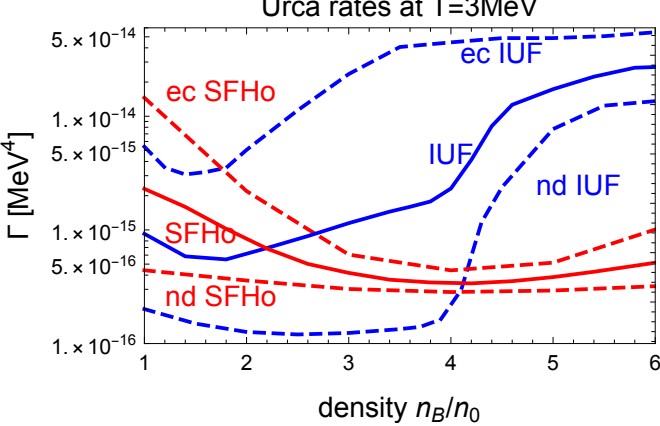

**Figure 8.** Urca (direct plus modified) rates for IUF and SFHo EoSs at $T = 3\,\mathrm{MeV}$. When $\Delta\mu = 0$ (dashed lines) the rates for neutron decay (nd) and electron capture (ec) do not balance. With the correct choice of $\Delta\mu$ (Figures 6 and 7) the neutron decay and electron capture rates (solid lines) become equal, and the system is in beta equilibrium.

For IUF, the mismatch between electron capture and neutron decay is greatest just below the IUF direct Urca threshold density of $4\,n_0$, which explains why for IUF $\Delta\mu$ reaches its highest value there (Figure 6). For SFHo, the mismatch is smallest at that density, which explains why for SFHo $\Delta\mu$ reaches a local minimum there (Figure 7).

Figures 9 and 10 give further insight into the density dependence of the rates by showing the separate contributions from direct and modified Urca.

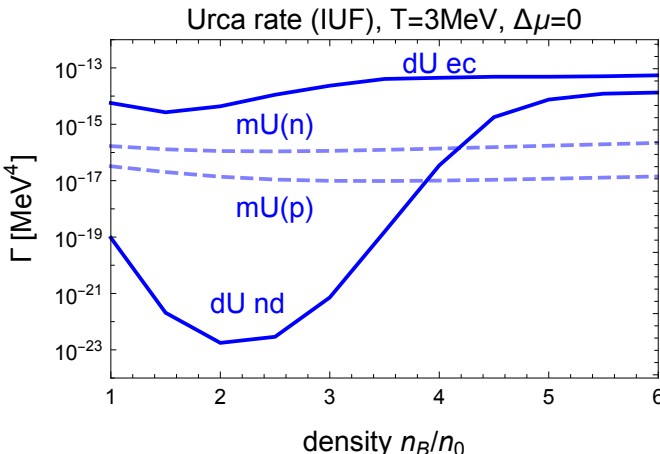

**Figure 9.** Urca rates calculated using the IUF EoS at $T = 3\,\text{MeV}$. Because $\Delta\mu = 0$ there is a large mismatch between the direct Urca rates for neutron decay and electron capture. Modified Urca (with neutron spectator (n) and proton spectator (p)) rates are calculated in the Fermi surface approximation and therefore match automatically.

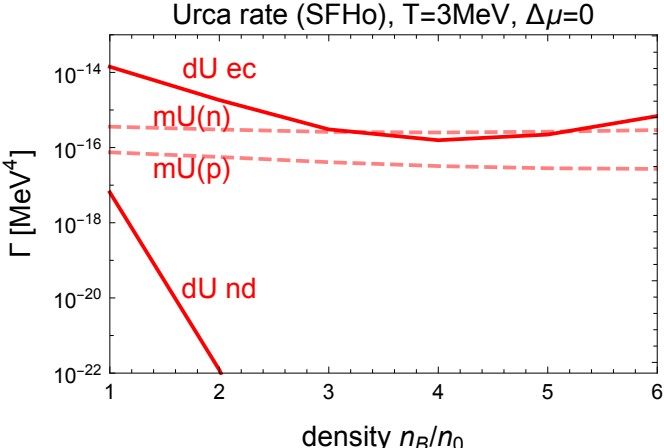

**Figure 10.** Urca rates calculated using the SFHo EoS at $T = 3\,\text{MeV}$. Because $\Delta\mu = 0$ there is a large mismatch between the direct Urca rates for neutron decay and electron capture. Modified Urca (with neutron spectator (n) and proton spectator (p)) rates are calculated in the Fermi surface approximation and therefore match automatically.

For IUF (Figure 9), in the dUrca-forbidden density range one would expect that the direct Urca rates should be exponentially suppressed at low temperature, leaving the modified Urca rates which automatically balance when $\Delta\mu = 0$ because they are calculated in the Fermi surface approximation. We see that the direct Urca neutron capture rate is indeed strongly suppressed, but the direct Urca electron capture rate only shows a slight reduction below the threshold, and remains well above the modified Urca rates. This mismatch is what leads to a positive correction $\Delta\mu$ in beta equilibrium. We will explain below why this is the case.

For SFHo (Figure 10), the analysis is similar: neutron decay is heavily suppressed as expected in the dUrca-forbidden region (up to infinite density), but electron capture is much less suppressed. In the middle density range (3 to 5 $n_0$) where mUrca is dominant there is no need for a correction, since the mUrca rates balance at $\Delta\mu = 0$. However, at lower or higher densities the direct Urca electron capture rate becomes large enough to dominate, so a positive $\Delta\mu$ will be required to pull it down and establish equilibrium between neutron decay and electron capture.

In the next subsection we analyze the imbalance between electron capture and neutron decay rates in the dUrca-forbidden density range. This imbalance is the reason a nonzero

$\Delta\mu$ is required in beta equilibrium. We can understand the difference in the rates, and their density dependence, by looking at which parts of the phase space dominate the rate integrals. This is largely determined by the Fermi-Dirac factors in the rate integrals, since the matrix element depends only weakly on the magnitudes of the momenta.

### 6.3. Direct Urca Suppression Factors

The density and temperature dependence of the direct Urca rates is dominated by the Fermi-Dirac factors. Below the dUrca threshold density, at zero temperature all direct Urca processes would be forbidden, but at nonzero temperature the Fermi surfaces are blurred, so there is some nonzero occupation of particle and hole states in regions of momentum space where the direct Urca process is kinematically allowed. The rate is governed by the Fermi-Dirac suppression factors for those momentum states.

At each density and temperature we search for the combination of momenta that is least suppressed, i.e., that maximizes the product of Fermi-Dirac factors in the rate integral while maintaining energy-momentum conservation. The magnitude of that product of Fermi-Dirac factors tells us how suppressed the whole process will be, at that density and temperature.

Below the direct Urca threshold density, considering particles near their Fermi surfaces, the neutron has a momentum larger than the sum of proton and electron momenta, even if the proton and electron are coaligned (see Figure 1). In this regime, the direct Urca kinematics will become essentially one-dimensional, as this is how the electron and proton momenta can come closest to adding up to the large neutron momentum. We take the neutron momentum to be positive, so a negative momentum indicates motion in the direction opposite of the neutron. For momentum conservation to hold, the electron and proton will have to be away from their Fermi surfaces. In the assumption of one-dimensional kinematics, we determine the optimal momenta $\{k_n^{\text{opt}}, k_p^{\text{opt}}, k_e^{\text{opt}}, k_\nu^{\text{opt}}\}$ as follows. For neutron decay, we maximize $f_n(1-f_p)(1-f_e)$ and for electron capture we maximize $(1-f_n)f_pf_e$. Energy and (one-dimensional) momentum conservation impose two constraints on the momentum, leaving two independent momenta over which to maximize.

The results of this maximization exercise are shown for the IUF EoS in Figures 11 and 12, and for SFHo in Figures 13 and 14.

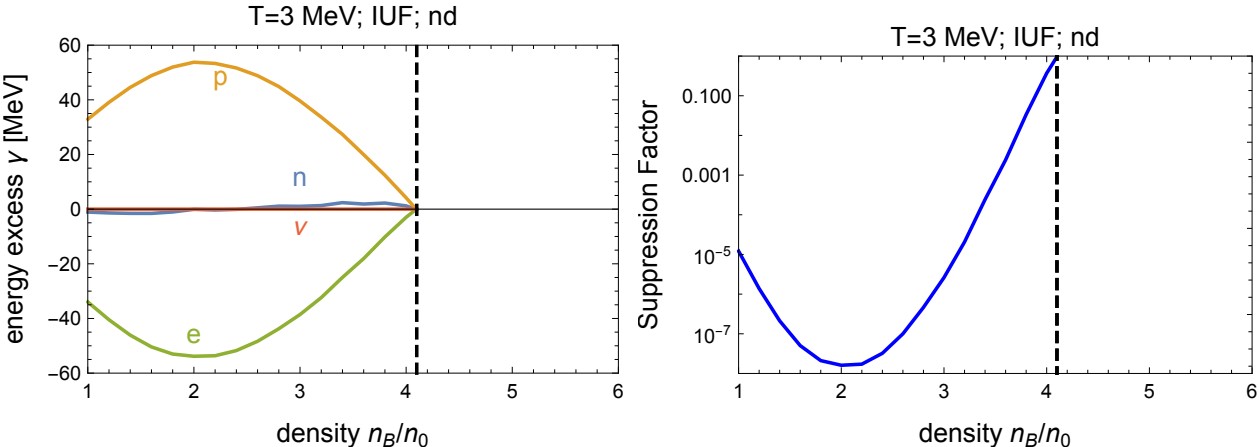

**Figure 11.** The optimal kinematics for neutron decay for the IUF EoS. **Left panel**: the least suppressed kinematic arrangement, showing the energy distance $\gamma$ of each particle from its Fermi surface. **Right panel**: the Fermi-Dirac suppression factor, $e^{-|\gamma_e|/T}e^{-|\gamma_n|\Theta(\gamma_n)/T}$ which is dominated by the difficulty of finding an electron hole at energy $\gamma_e$ below its Fermi surface.

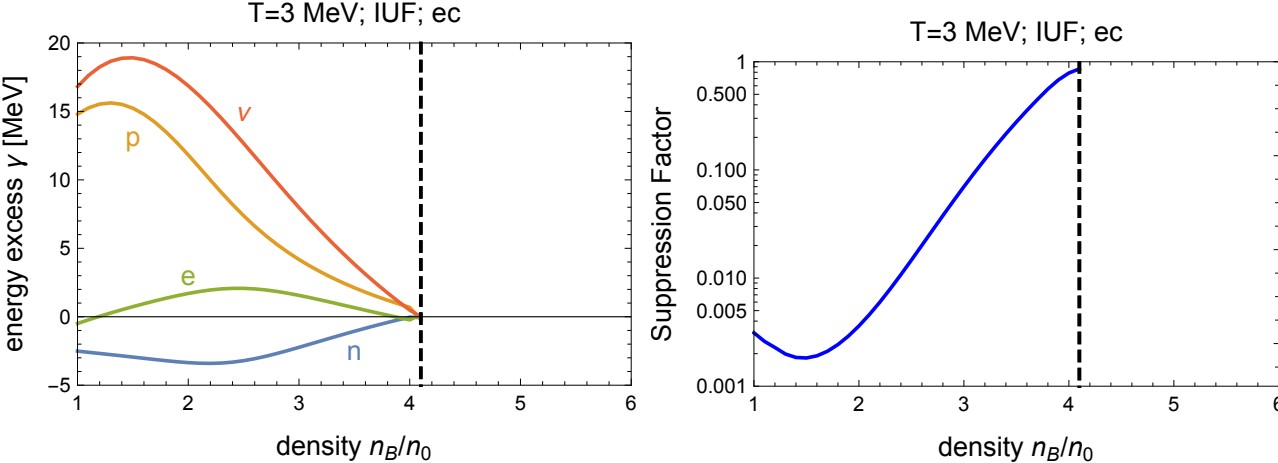

**Figure 12.** The optimal kinematics for electron capture for the IUF EoS. **Left panel**: the least suppressed kinematic arrangement, showing the energy distance $\gamma$ of each particle from its Fermi surface. **Right panel**: the overall Fermi-Dirac suppression factor, $e^{-|\gamma_p|/T}e^{-|\gamma_e|\Theta(\gamma_e)/T}e^{-|\gamma_n|\Theta(-\gamma_n)/T}$, which is dominated by the difficulty of finding a proton at energy $\gamma_p$ above its Fermi surface.

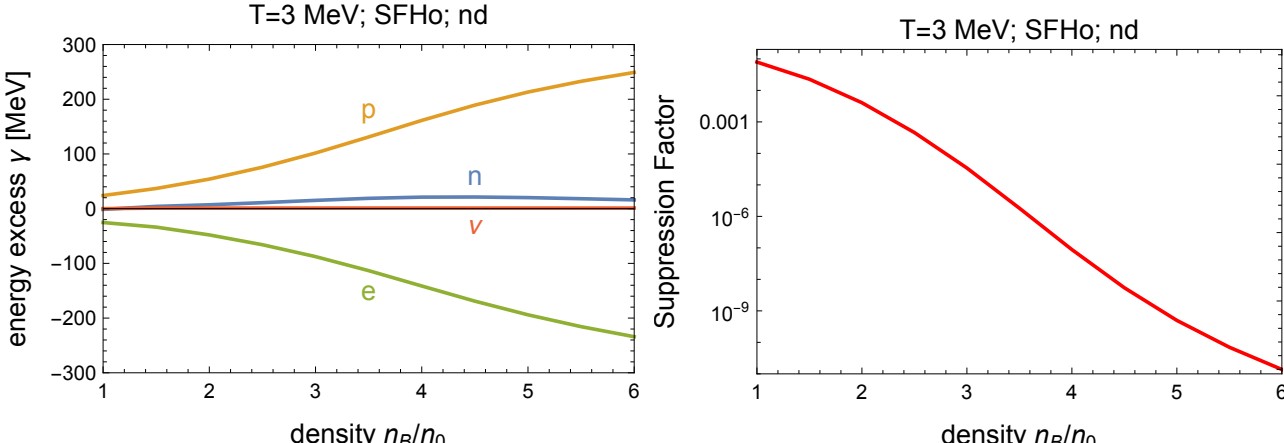

**Figure 13.** The optimal kinematics for neutron decay at $T = 3\,\text{MeV}$ for SFHo, obtained by maximizing the Fermi-Dirac products. The suppression factor, $e^{-|\gamma_e|/T}e^{-|\gamma_n|\Theta(\gamma_n)/T}$ is dominated by the difficulty of finding an electron hole below its Fermi surface.

The left panels show how far from their Fermi surfaces the particles are in the least Fermi-Dirac-suppressed kinematic configuration. For each particle $i$ we show $\gamma_i \equiv E_i^{\text{opt}} - E_{Fi}$, which is the extra energy the particle with its optimal momentum has relative to its Fermi energy. The curves only exist in the dUrca-forbidden region, which for IUF ends at $4.1\,n_0$. (In the dUrca-allowed region all particles can be on their Fermi surfaces, so the curves would be trivially zero and are not shown). The right panels show the maximum value of the Fermi-Dirac factor, which gives the overall suppression of the process.

### 6.3.1. Neutron Decay

Direct Urca neutron decay is suppressed because the neutrons at their Fermi surface have just enough energy to make a proton and electron near their Fermi surfaces (this is a consequence of the beta equilibrium condition (6)), but too much momentum (Figure 1). The process can still proceed (with an exponential suppression factor) by exploiting the thermal blurring of the Fermi surfaces. Figure 11 (IUF) and Figure 13 (SFHo) show that the best option is to create a proton at energy $\gamma_p$ above its Fermi surface and an electron at energy $\gamma_e = -\gamma_p$ which is below its Fermi surface. The co-linear proton and electron now have more momentum then when they were both on their Fermi surfaces because the

proton's momentum rises rapidly with $\gamma_p$ because the proton is less relativistic, whereas the electron's momentum drops more slowly as $\gamma_e$ becomes more negative, because the electron is ultrarelativistic. The creation of the proton incurs no Fermi-Dirac suppression because states above the Fermi surface are mostly empty, but the creation of the electron is suppressed by a Fermi-Dirac factor of $e^{-|\gamma_e|/T}$ reflecting the scarcity of electron holes available to take such an electron. The net suppression of the rate, $e^{-|\gamma_e|/T}e^{-|\gamma_n|\Theta(\gamma_n)/T}$, is shown in the right panels of Figure 11 (IUF) and Figure 13 (SFHo). For IUF we see the strongest suppression at around $2\,n_0$, which explains the density dependence of the IUF neutron decay rate shown in Figure 9. For SFHo, since the dUrca-forbidden region extends up to infinite density, and the momentum deficit remains large across the density range surveyed, we see stronger suppression that does not relent at the upper end of the density range, explaining the almost total suppression seen in Figure 10.

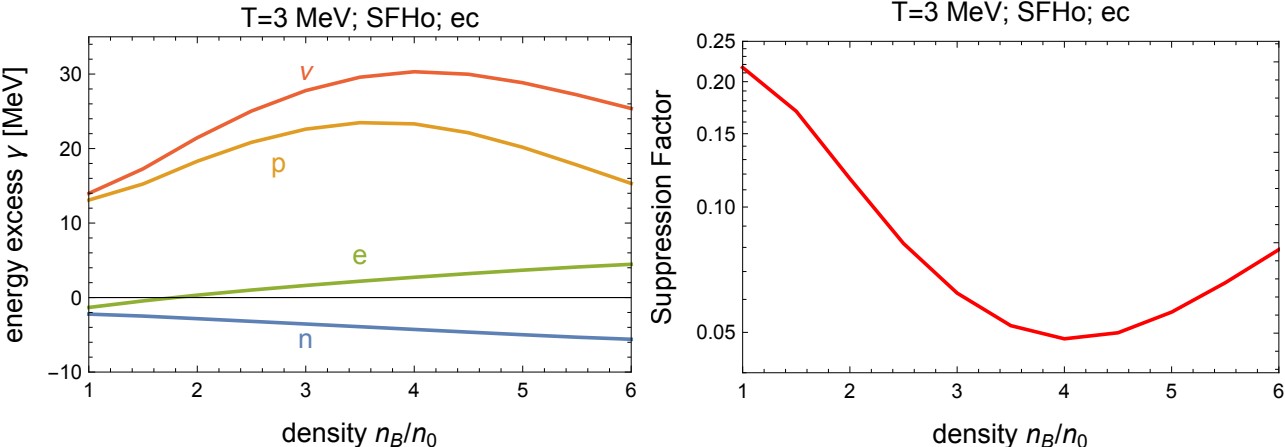

**Figure 14.** The optimal kinematics for electron capture at $T = 3\,\text{MeV}$ for SFHo, obtained by maximizing the Fermi-Dirac products. The suppression factor, $e^{-|\gamma_p|/T}e^{-|\gamma_e|\Theta(\gamma_e)/T}e^{-|\gamma_n|\Theta(-\gamma_n)/T}$, is dominated by the difficulty of finding a proton above its Fermi surface.

We can understand the density dependence of $\gamma_p$ in terms of the one-dimensional model within which the maximization was performed.

We assume that as seen in Figure 11 (IUF) and Figure 13 (SFHo), the neutron remains on its Fermi surface, and the neutrino takes negligible energy/momentum, since lack of momentum to build the final state is the main obstacle. Conservation of energy and momentum then tells us that

$$k_{Fn} = k_p^{\text{opt}} + k_e^{\text{opt}}, \tag{25}$$

$$E_{Fn} = E_p(k_p^{\text{opt}}) + k_e^{\text{opt}}. \tag{26}$$

Using the dispersion relations (8) we can solve for $k_p^{\text{opt}}$ and $k_e^{\text{opt}}$ and, after using that $E_{Fn} = E_{Fp} + E_{Fe}$ (since we have assumed $\Delta\mu = 0$), we find

$$k_p^{\text{opt}} - k_{Fp} = \frac{\Delta k(2E_{Fp}^* - \Delta k)}{2(E_{Fp}^* + k_{Fp} - k_{Fn})}, \tag{27}$$

where $\Delta k \equiv k_{Fn} - k_{Fp} - k_{Fe}$ is the momentum deficit (we plotted the surplus $-\Delta k$ in Figure 1). From this analysis, we learn that the density dependence of $\gamma_p$, and therefore the rate, not only depends on the momentum deficit $\Delta k$, but on the relative behavior of the neutron and proton Fermi momenta and their effective masses.

Although the momentum deficit $\Delta k$ in IUF monotonically shrinks with density, $\gamma_p$ shows a slight increase at low densities due to the fast drop of the effective proton mass $m_p^*$ (see Figure 3). This fast decrease counter-intuitively leads $E_{Fp}^*$ to drop with density,

while the real Fermi energy, which includes the nuclear mean field, $U_p$, rises with density as expected. Closer to the threshold density, the momentum deficit dominates the behavior of $\gamma_p$ and the rate, so that $\gamma_p$ goes to zero at the threshold as $\Delta k$ approaches zero, leading to $k_p^{\text{opt}} = k_{Fp}$ as expected.

For the SFHo EoS, the direct Urca momentum deficit is only varying weakly with density (see again Figure 1). Although the momentum deficit is slowly falling, $\gamma_p$ continues to rise with density as shown in Figure 13. This is due to the neutron Fermi momentum which rises fast enough that the denominator in Equation (27) decreases by more than a factor of five in the studied density range while the momentum surplus stays nearly constant in comparison.

### 6.3.2. Electron Capture

In the dUrca-forbidden density range, using the one-dimensional kinematics described above, we find that the optimal kinematics for electron capture has a proton above its Fermi surface and an electron close to its Fermi surface combining to make a neutron slightly below its Fermi surface and a neutrino. The Fermi-Dirac suppression factor is $e^{-\gamma_p/T}e^{-|\gamma_e|\Theta(\gamma_e)/T}e^{-|\gamma_n|\Theta(-\gamma_n)/T}$, reflecting the scarcity of protons and electrons above their Fermi surfaces, and of neutron holes below the neutron Fermi surface.

Figures 12 and 14 show the corresponding energy excesses $\gamma_i$ and Fermi-Dirac suppression factors. In the right panels we see that in the dUrca-forbidden region, electron capture is somewhat suppressed but not nearly as suppressed as neutron decay. This is because, as we explain below, it is able to proceed using a proton that is much closer to its Fermi surface than is possible for neutron decay, and there is correspondingly less Fermi-Dirac suppression (compare the left panels of Figure 11 vs. Figure 12, and Figure 13 vs. Figure 14).

The special feature of electron capture is that there is a very efficient way to exploit the thermal blurring of the Fermi surfaces. Given a momentum shortfall $\Delta k \equiv k_{Fn} - k_{Fp} - k_{Fe}$, we can start with a proton whose momentum is less than $\Delta k$ above the Fermi surface. The rarity of finding such a proton leads to a Fermi-Dirac suppression factor of $e^{-|\gamma_p|/T}$. This proton captures an electron near its Fermi surface with momentum parallel to the proton's. At this point their combined momentum is not enough to make a neutron on its Fermi surface, and there is excess energy. However, we can use that excess energy to create, along with a neutron on its Fermi surface, a neutrino whose momentum partly cancels the neutron momentum, so the combined momentum of the proton and electron is enough to create that final state.

Because of the "help" from the neutrino, the proton does not need to be as far above its Fermi surface as the proton in neutron decay, so the electron capture rate is suppressed by a smaller Fermi-Dirac factor,

The density dependence of the suppression factors (right panels of Figure 12 for IUF and Figure 14 for SFHo) explain the density dependence of the direct Urca electron capture rates shown in Figures 9 and 10.

To understand the density dependence of $\gamma_p$, we can perform a similar analysis as for neutron decay. We now assume neutron and electron to be on their Fermi surfaces, as shown in Figure 12 (IUF) and Figure 14 (SFHo), which is not as good as an assumption compared to the neutron decay analysis, but still helps us to gain insight into the behavior of the rates. Energy-momentum conservation again allows us to deduce that

$$k_{Fn} = k_p^{\text{opt}} + k_{Fe} + k_\nu^{\text{opt}}, \tag{28}$$

$$E_{Fn} + k_\nu^{\text{opt}} = E_p(k_p^{\text{opt}}) + k_{Fe}^{\text{opt}}, \tag{29}$$

which leads, following the same procedure as in the neutron decay case, to

$$k_p^{\text{opt}} - k_{Fp} = \frac{\Delta k(\Delta k + 2E_{Fp}^*)}{2(E_{Fp}^* + k_{Fn} - k_{Fp})}. \tag{30}$$

For IUF at low densities, we can neglect the proton Fermi momentum compared to the effective mass. The behavior of $\gamma_p$ is then again dominated by the effective proton mass, whose rapid decrease overcomes the rising neutron Fermi momentum at low densities. This pushes the proton further away from its Fermi surface at low densities, before the momentum surplus dominates the behavior of $\gamma_p$ as the threshold is approached. As for neutron decay, $\Delta k = 0$ at the threshold, therefore the rate is again dominated by particles on their respective Fermi surfaces.

For SFHo, the momentum surplus is becoming smaller from $n_0$ to $3\,n_0$ while the combination of the effective masses and Fermi momenta in (30) varies slowly with density. This allows the behavior of the momentum surplus $\Delta k$ to dominate the behavior of $\gamma_p$ at low densities, so both are increasing and therefore pushing the proton further away from its Fermi surface initially. At higher densities, SFHo is seemingly approaching asymptotically a direct Urca threshold. Both the momentum surplus and the Fermi momenta and effective masses in Equation (30) are pushing the ideal proton momentum back closer to the Fermi surface. Overall, the behavior of the electron capture rate in SFHo can therefore largely be explained by the density dependence of the momentum surplus.

*6.4. Nonrelativistic Rate vs. Relativistic Rate*

In Section 3 we emphasized that as the density rises above about $2n_0$ relativistic corrections become important in the nucleon dispersion relations. In this section, we illustrate the importance of relativistic corrections in the neutron decay rate.

6.4.1. Direct Urca Neutron Decay

Figure 4 shows various approximations to the direct Urca neutron decay rate at $T = 3\,\mathrm{MeV}$ (with $\Delta\mu = 0$). We show the rate calculated with fully relativistic dispersion relations, with the nonrelativistic dispersion relation

$$E_N = m_N^* + \frac{p_N^2}{2m_N^*} + U_N, \tag{31}$$

and with the "vacuum dispersion relation" used in [5],

$$E_N = m_{\mathrm{eff,N}} + \frac{p_N^2}{2m_N}, \tag{32}$$

where $m_N = 940\,\mathrm{MeV}$, and $m_{\mathrm{eff,N}}$ is chosen such that $E_N(p_F) = \mu_N$.

For the nonrelativistic curves, we use a corresponding nonrelativistic approximation of the rescaled matrix element (12),

$$\mathcal{M} = 1 + 3g_A^2 + (1 - g_A^2)\frac{\vec{p}_e \cdot \vec{p}_\nu}{E_e E_\nu}, \tag{33}$$

see Refs. [5,31], and the derivation in Appendix C of [60]. We see that relativistic corrections make an enormous difference to the rate. The nonrelativistic approximation is reasonably accurate at low density (where the nucleons are indeed nonrelativistic) but overestimates the rate by up to eight orders of magnitude (at $T = 3\,\mathrm{MeV}$) between $2\,n_0$ and the direct Urca threshold at $4.1\,n_0$. Due to the breakdown of the nonrelativistic approximation, the direct Urca threshold condition is incorrectly already fulfilled below two times saturation density, which explains the steep increase of the nonrelativistic rate around this density. For a detailed discussion of the density dependence of the relativistic rate, see Section 6.3.

The thermal blurring of the Fermi energy, which is proportional to the temperature $T$, translates to a blurring in momentum space of order $T/v_F$, where $v_F$ is the Fermi velocity. In the correct relativistic treatment, $v_F$ has an upper bound of 1, whereas for the nonrelativistic dispersion relation, the Fermi velocity grows without a limit. This leads to a suppression of the nonrelativistic rate at higher densities which partially cancels the effects of the earlier threshold.

The "vacuum dispersion relation" gives a rate that is one to eight orders of magnitude too large (at $T = 3\,\text{MeV}$), and is less suppressed at higher densities since the corresponding Fermi velocity stays comparatively small in the plotted density range.

### 6.4.2. Modified Urca Neutron Decay

Figure 5 shows the importance of using relativistic dispersion relations in calculating modified Urca. The rates are calculated for the IUF equation of state in the Fermi surface approximation at $T = 3\,\text{MeV}$. The relativistic rate is about 1 to 2 orders of magnitude smaller than the nonrelativistic rate. The modified Urca rates are not sensitive to the direct Urca threshold because of the spectator providing extra momenta. Much of the difference between the nonrelativistic calculation and the relativistic calculation comes from the prefactors, as shown in Section 5 and Equations (57), (58), (60) and (61). The relativistic rates are suppressed by $\prod_i m_i^* / E_i^*$, where $i$ is the index for each of the nucleons participating the interaction. Notice that the proton-spectator modified Urca rate is always less than the neutron-spectator rate because the proton Fermi surface, and its accompanying phase space, is smaller.

## 7. Conclusions

We have investigated the conditions for beta equilibrium in nuclear matter in neutron stars, focusing on the temperature range where the material is cool enough so that neutrinos escape ($T \lesssim 5\,\text{MeV}$) but warm enough so that nonzero-temperature corrections to the Fermi surface approximation play an important role ($T \gtrsim 1\,\text{MeV}$).

Previous work [5] found that a nonzero-temperature correction $\Delta\mu$ to the traditional beta equilibrium condition (Equation (7)) was required to balance the rate of neutron decay against the rate of electron capture. We have improved on that calculation using a consistent description of nuclear matter, based on two relativistic mean field models, IUF and SFHo.

We find that when using relativistic mean field models it is important to use the full relativistic dispersion relations of the nucleons. In these theories the effective masses drop quickly with density, so the neutrons become relativistic at densities of 2 to $3\,n_0$. Using nonrelativistic nucleon dispersion relations can make the modified Urca rates wrong by an order of magnitude and the direct Urca rates wrong by many orders of magnitude.

Our results for the nonzero-temperature correction $\Delta\mu$ are shown in Figures 6 and 7. We find that it rises with the temperature, and can be of order 10 to 20 MeV for temperatures in the 3 to 5 MeV range. The density dependence is quite different for the two EoSs that we studied, and we showed in detail how it depends on specific properties of the EoS.

We find that the nonzero-temperature correction plays an important role in the correct calculation of Urca rates. Using the naive (low-temperature) beta equilibrium condition $\mu_n = \mu_p + \mu_e$ at $T = 3\,\text{MeV}$ would yield electron capture rates that are too large by an order of magnitude, and neutron decay rates that are too small by an order of magnitude (Figure 8). This would significantly affect calculations of neutrino emissivity in the cooler regions of a neutron star merger, and therefore the estimated energy loss due to neutrinos. Currently used neutrino leakage schemes (e.g., Ref. [61] and references therein), which often treat the temperature range $T \lesssim 5\,\text{MeV}$ as neutrino free streaming, need to be adapted to the corrected beta equilibrium. Additionally, the bulk viscosity of nuclear matter [62] depends on the rate of the Urca process which restores the system to beta equilibrium. The improved calculation of the Urca rates presented here will modify the temperatures and densities at which bulk viscosity reaches its maximum strength. Using the correct beta equilibrium condition also affects the equation of state: a recent study estimated its impact to be at the 5% level [63], and it would be interesting to evaluate the impact by performing a merger simulation using an EoS that incorporates the finite-temperature correction described in this paper.

**Author Contributions:** Conceptualization, M.G.A., A.H., S.P.H.; methodology, all authors; formal analysis, all authors; writing, all authors; visualization, Z.Z. All authors have read and agreed to the published version of the manuscript.

**Funding:** M.G.A., A.H. and Z.Z. are partly supported by the U.S. Department of Energy, Office of Science, Office of Nuclear Physics, under Award No. #DE-FG02-05ER41375. Z.Z. received additional support from the McDonnell Center for the Space Sciences at Washington University. S.P.H. is supported by the U. S. Department of Energy grant DE-FG02-00ER41132 as well as the National Science Foundation grant No. PHY-1430152 (JINA Center for the Evolution of the Elements).

**Institutional Review Board Statement:** No application.

**Informed Consent Statement:** No application.

**Data Availability Statement:** No application.

**Acknowledgments:** We thank A. Steiner for useful discussions.

**Conflicts of Interest:** The authors declare no conflict of interest.

**Appendix A. The SFHo Relativistic Mean Field Theory**

The Lagrangian for the SFHo relativistic mean field model is given in Refs. [42,64] and reads

$$\mathcal{L} = \mathcal{L}_N + \mathcal{L}_M + \mathcal{L}_l, \tag{A1}$$

$$\mathcal{L}_N = \bar{\psi}(i\gamma^\mu\partial_\mu - m_N + g_\sigma\sigma - g_\omega\gamma^\mu\omega_\mu - \frac{g_\rho}{2}\boldsymbol{\tau}\cdot\boldsymbol{\rho}_\mu\gamma^\mu)\psi, \tag{A2}$$

with bold symbols being vectors in iso-space, $\boldsymbol{\tau}$ being the iso-spin generators, and

$$\begin{aligned}
\mathcal{L}_M =& \frac{1}{2}\partial_\mu\sigma\partial^\mu\sigma - \frac{1}{2}m_\sigma^2\sigma^2 - \frac{bM}{3}(g_\sigma\sigma)^3 - \frac{c}{4}(g_\sigma\sigma)^4 - \frac{1}{4}\omega_{\mu\nu}\omega^{\mu\nu} + \frac{1}{2}m_\omega^2\omega_\mu\omega^\mu \\
&+ \frac{\zeta}{24}g_\omega^4(\omega_\mu\omega^\mu)^2 - \frac{1}{4}\boldsymbol{B}_{\mu\nu}\cdot\boldsymbol{B}^{\mu\nu} + \frac{1}{2}m_\rho^2\boldsymbol{\rho}_\mu\cdot\boldsymbol{\rho}^\mu + \frac{\xi}{24}g_\rho^4(\boldsymbol{\rho}_\mu\cdot\boldsymbol{\rho}^\mu)^2 \\
&+ g_\rho^2\Big[\sum_{i=1}^{6}a_i\sigma^i + \sum_{j=1}^{3}b_j(\omega_\mu\omega^\mu)^j\Big]\boldsymbol{\rho}_\mu\cdot\boldsymbol{\rho}^\mu, 
\end{aligned} \tag{A3}$$

where

$$\omega_{\mu\nu} = \partial_\mu\omega_\nu - \partial_\nu\omega_\mu, \tag{A4}$$

$$\boldsymbol{B}_{\mu\nu} = \partial_\mu\boldsymbol{\rho}_\nu - \partial_\nu\boldsymbol{\rho}_\mu. \tag{A5}$$

The lepton contribution

$$\mathcal{L}_l = \bar{\psi}_e\left(i\gamma^\mu\partial_\mu - m_e\right)\psi_e, \tag{A6}$$

consists of free electrons with a mass of $m_e = 0.511\,\text{MeV}$. In our calculations we use the values of the masses and couplings given in the online CompOSE database. These are listed in Table A1. In the table,

$$c_\sigma = g_\sigma/m_\sigma, \tag{A7}$$

$$c_\omega = g_\omega/m_\omega, \tag{A8}$$

$$c_\rho = g_\rho/m_\rho. \tag{A9}$$

**Table A1.** SFHo parameter values taken from CompOSE (https://compose.obspm.fr/eos/34, accessed on 27 April 2021). The last three masses are taken from [42].

| Quantity | Unit | Value |
|---|---|---|
| $c_\sigma$ | fm | 3.1791606374 |
| $c_\omega$ | fm | 2.2752188529 |
| $c_\rho$ | fm | 2.4062374629 |
| $b$ | | $7.3536466626 \times 10^{-3}$ |
| $c$ | | $-3.8202821956 \times 10^{-3}$ |
| $\zeta$ | | $-1.6155896062 \times 10^{-3}$ |
| $\tilde{\zeta}$ | | $4.1286242877 \times 10^{-3}$ |
| $a_1$ | fm$^{-1}$ | $-1.9308602647 \times 10^{-1}$ |
| $a_2$ | | $5.6150318121 \times 10^{-1}$ |
| $a_3$ | fm | $2.8617603774 \times 10^{-1}$ |
| $a_4$ | fm$^2$ | 2.7717729776 |
| $a_5$ | fm$^3$ | 1.2307286924 |
| $a_6$ | fm$^4$ | $6.1480060734 \times 10^{-1}$ |
| $b_1$ | | 5.5118461115 |
| $b_2$ | fm$^2$ | $-1.8007283681$ |
| $b_3$ | fm$^4$ | $4.2610479708 \times 10^2$ |
| $m_\sigma$ | fm$^{-1}$ | 2.3689528914 |
| $m_\omega$ | fm$^{-1}$ | 3.9655047020 |
| $m_\rho$ | fm$^{-1}$ | 3.8666788766 |
| $m_n$ | MeV | 939.565346 |
| $m_p$ | MeV | 938.272013 |
| $M$ | MeV | 939 |

**Appendix B. Direct Urca Neutron Decay Rate**

From Fermi's Golden rule, we have the rate Equation (9) [31,57]

$$\Gamma_{\text{nd}} = \int \frac{d^3 k_n}{(2\pi)^3} \frac{d^3 k_p}{(2\pi)^3} \frac{d^3 k_e}{(2\pi)^3} \frac{d^3 k_\nu}{(2\pi)^3} \frac{\sum |M|^2}{(2E_n^*)(2E_p^*)(2E_e)(2E_\nu)} (2\pi)^4 \delta^{(4)}(k_n - k_p - k_e - k_\nu)$$
$$f_n(1 - f_p)(1 - f_e). \tag{A10}$$

There is no neutrino Fermi-Dirac factor because we assume the medium is neutrino-transparent, i.e., neutrinos escape the star. The spin-summed matrix element is [11]

$$\sum |M|^2 = 32G^2[(g_A^2 - 1)m_n^* m_p^*(k_e \cdot k_\nu) + (g_A - 1)^2(k_e \cdot k_n)(k_p \cdot k_\nu)$$
$$+ (1 + g_A)^2(k_p \cdot k_e)(k_n \cdot k_\nu)], \tag{A11}$$

where $G = G_F \cos \theta_c$, $G_F = 1.166 \times 10^{-11}$ MeV$^{-2}$ is the Fermi constant and $\theta_c = 13.04°$ is the Cabbibo angle. As they originate from spin summations (see Appendix B of [9]), the 4-vector dot products in the matrix element (A11) are $k^\mu = (E^*, \mathbf{k})$.

It is convenient to define the rescaled dimensionless matrix element

$$\mathcal{M} \equiv \frac{\sum |M|^2}{32G^2 E_n^* E_p^* E_e E_\nu} \tag{12}$$
$$= \frac{(g_A^2 - 1)m_n^* m_p^*(k_e \cdot k_\nu) + (g_A - 1)^2(k_e \cdot k_n)(k_p \cdot k_\nu) + (1 + g_A)^2(k_p \cdot k_e)(k_n \cdot k_\nu)}{E_n^* E_p^* E_e E_\nu}.$$

In the nonrelativistic limit, since $g_A \approx 1$, $\mathcal{M} \approx (1 + 3g_A^2) \sim 4$ [11,20,31,34,65,66].

The neutron decay rate can now be written

$$\Gamma_{\text{nd}} = \frac{2G^2}{(2\pi)^8} \int d^3k_n d^3k_p d^3k_e d^3k_\nu \, \mathcal{M} \, \delta^{(4)}(k_n - k_p - k_e - k_\nu) f_n(1 - f_p)(1 - f_e). \quad (13)$$

The 12-dimensional integral can be reduced to a 5-dimensional integral as follows. Integrating over the 3-momentum conservation delta functions reduces the integral to 9 dimensions (compare (E.1) in Ref. [60])

$$\Gamma_{\text{nd}} = \frac{2G^2}{(2\pi)^8} \int d^3k_n d^3k_p d^3k_e \, \mathcal{M} \delta(E_n - E_p - E_e - |\vec{k}_n - \vec{k}_p - \vec{k}_e|) f_n(1 - f_p)(1 - f_e) \, . \quad (14)$$

The remaining delta function imposes energy conservation in the creation of the neutrino: $E_\nu = |\vec{k}_\nu|$, so the argument of the delta function is

$$g(\phi) \equiv E_\nu - |\vec{k}_n - \vec{k}_p - \vec{k}_e| \, , \quad (15)$$
$$E_\nu \equiv E_n - E_p - E_e \, .$$

Each momentum integral can be written in polar co-ordinates as $d^3k = k^2 dk \, dz \, d\phi$ where $z = \cos\theta$. Setting up the following coordinate system (see Appendix E in [60])

$$\vec{k}_n = k_n(0, 0, 1) \, , \quad (16)$$
$$\vec{k}_p = k_p(\sqrt{1 - z_p^2}, 0, z_p) \, , \quad (17)$$
$$\vec{k}_e = k_e(\sqrt{1 - z_e^2} \cos\phi, \sqrt{1 - z_e^2} \sin\phi, z_e) \, , \quad (18)$$

allows us to integrate over $z_n$ and $\phi_n$ yielding a factor of $4\pi$ and over $\phi_p$ yielding a factor of $2\pi$, which eliminates three angular integrals, so that (compare (E.5) in [60])

$$\Gamma_{\text{nd}} = \frac{G^2}{16\pi^6} \int_0^\infty dk_n \int_0^{k_p^{\max}} dk_p \int_0^{k_e^{\max}} dk_e k_n^2 k_p^2 k_e^2 \, f_n(1 - f_p)(1 - f_e) \, I(k_n, k_p, k_e) \, , \quad (19)$$

where

$$I(k_n, k_p, k_e) \equiv \Theta(E_\nu) \int_{-1}^1 dz_p \int_{-1}^1 dz_e \int_0^{2\pi} d\phi \, \mathcal{M} \, \delta(g(\phi)) \, . \quad (20)$$

Please note that for simplicity we label the electron azimuthal angle as $\phi$ (rather than $\phi_e$). The factor of $\Theta(E_\nu)$ restricts the integral to the region of momentum space where the neutrino energy $E_\nu(k_n, k_p, k_e)$ is positive, which is a requirement for the emission of a neutrino. This condition leads to the upper limits on the proton and electron momenta. If we perform the integrals in the order shown in (19) then the electron momentum integral is the inner integral, so it is performed for known values of $k_n$ and $k_p$, so the constraint $E_\nu > 0$ corresponds to $E_e < E_n - E_p$. Similarly, the $k_p$ integral is performed for a known value of $k_n$, so its range is constrained by requiring that there be enough energy to create an electron (of unknown momentum) and a neutrino, $E_p < E_n - m_e$. This leads to upper limits on the proton and electron integral,

$$k_p^{\max} = \Theta(E_n - U_p - m_p - m_e)\sqrt{(E_n - U_p - m_e)^2 - m_p^2} \, , \quad (21)$$
$$k_e^{\max} = \Theta(E_n - E_p - m_e)\sqrt{(E_n - E_p)^2 - m_e^2} \, . \quad (22)$$

In the delta function in Equation (20),

$$g(\phi) = E_\nu - \sqrt{R + S \cos \phi}, \tag{23}$$

$$\text{where} \quad R \equiv k_n^2 + k_p^2 + k_e^2 - 2k_n k_e z_e - 2k_n k_p z_p + 2k_p k_e z_p z_e, \tag{24}$$

$$S \equiv 2k_p k_e \sqrt{1 - z_p^2} \sqrt{1 - z_e^2}. \tag{25}$$

Since $g(\phi)$ depends on $\phi$ only via $\cos \phi$ there will be either zero or two solutions to $g(\phi) = 0$, so

$$I(k_n, k_p, k_e) = 2\,\Theta(E_\nu) \int_{-1}^{1} dz_p \int_{-1}^{1} dz_e\, \Theta\big(S - |E_\nu^2 - R|\big) \frac{\mathcal{M}_{\phi_0}}{|g'(\phi_0)|}, \tag{26}$$

where $\mathcal{M}_{\phi_0}$ is the dimensionless rescaled matrix element (12) evaluated at $\phi_0$, which can be either of the two solutions of $g(\phi) = 0$,

$$\cos \phi_0 = \frac{E_\nu^2 - R}{S}. \tag{27}$$

It does not matter which solution we use for $\phi_0$ because $g$ is a function of $\cos \phi$ and $\mathcal{M}$ depends only on $\cos \phi$ and $\sin^2 \phi$, so the integrand has the same value for both the solutions. The theta function $\Theta(S - |E_\nu^2 - R|)$ imposes the condition that there are two solutions (rather than none), by limiting the integral to the domain where $-1 < \cos \phi_0 < 1$.

We now use (23) and (27) to evaluate the integrand in (26).

First, the Jacobian of the delta function is

$$|g'(\phi_0)| = \frac{\sqrt{S^2 - (E_\nu^2 - R)^2}}{2E_\nu}. \tag{28}$$

Using (28) in (26),

$$I = 4E_\nu \Theta(E_\nu) \int_{-1}^{1} dz_p \int_{-1}^{1} dz_e\, \frac{\Theta\big(S - |E_\nu^2 - R|\big)}{\sqrt{S^2 - (E_\nu^2 - R)^2}}\, \mathcal{M}_{\phi_0}. \tag{29}$$

Secondly, substituting (27) in to (A11) gives the matrix element

$$\mathcal{M}_{\phi_0} = \frac{1}{2} \frac{(g_A - 1)^2 F_1 + (g_A + 1)^2 F_2 + (g_A^2 - 1)F_3}{E_n^* E_p^* E_e E_\nu}, \tag{30}$$

where

$$F_1 = \left(k_n^2 + k_e^2 - k_p^2 - 2E_p^* E_\nu - E_\nu^2 - 2k_n k_e z_e\right)\left(k_n k_e z_e - E_e E_n^*\right), \tag{31}$$

$$F_2 = \left(k_n^2 + k_p^2 + k_e^2 + 2E_p^* E_e - E_\nu^2 - 2k_n(k_p z_p + k_e z_e)\right)\left(E_n^* E_\nu + k_n(k_p z_p + k_e z_e - k_n)\right), \tag{32}$$

$$F_3 = m_n^{*2}\left(k_e^2 - k_n^2 - k_p^2 + 2E_e E_\nu + E_\nu^2 + 2k_n k_p z_p\right). \tag{33}$$

*Limits of Angular Integration*

To speed up the numerical evaluation of (29) we implement the theta function as limits on the range of integration over $z_p$ and $z_e$. The condition $S > |E_\nu^2 - R|$ can be written (using (24), (25)) as

$$|a + bz_e| < c\sqrt{1 - z_e^2} \,, \tag{34}$$

$$\text{where} \quad a \equiv q^2 - k_n^2 - k_p^2 - k_e^2 + 2k_n k_p z_p \,, \tag{35}$$

$$b \equiv 2k_e(k_n - k_p z_p) \,, \tag{36}$$

$$c \equiv 2k_e k_p \sqrt{1 - z_p^2} \,. \tag{37}$$

The inequality (34) is obeyed for $z_e^- < z_e < z_e^+$ where

$$z_e^\pm = \frac{-ab \pm c\sqrt{c^2 + b^2 - a^2}}{b^2 + c^2} \,. \tag{38}$$

Please note that if the roots are real then they are always within the physical range $z_e \in [-1, 1]$. We can therefore put bounds on $z_p$ by requiring that (38) has real roots,

$$c^2 + b^2 > a^2$$
$$\Rightarrow 2k_p E_\nu > |E_\nu^2 + k_e^2 - k_n^2 - k_p^2 + 2k_n k_p z_p| \,. \tag{39}$$

This means that $z_p^- < z_p < z_p^+$, where

$$z_p^\pm = \frac{k_n^2 + k_p^2 - k_e^2 - E_\nu^2 \pm 2k_e E_\nu}{2k_n k_p} \,. \tag{40}$$

In this case, however, these bounds are not necessarily within the physical range $z_p \in [-1, 1]$, so the true bounds on the $z_p$ integral are

$$[z_p^{\min}, z_p^{\max}] = [z_p^+, z_p^-] \cap [-1, 1] \,. \tag{41}$$

We can now write the angular integral as

$$I = 4E_\nu \Theta(E_\nu) \int_{z_p^{\min}}^{z_p^{\max}} dz_p \int_{z_e^-}^{z_e^+} dz_e \frac{\mathcal{M}_{\phi_0}}{\sqrt{S^2 - (E_\nu^2 - R)^2}} \,. \tag{42}$$

Using this in (19) we obtain

$$\Gamma_{\mathrm{nd}} = \frac{G^2}{16\pi^6} \int_0^\infty dk_n \int_0^{k_p^{\max}} dk_p \int_0^{k_e^{\max}} dk_e \, k_n^2 k_p^2 k_e^2 f_n (1 - f_p)(1 - f_e)$$

$$\Theta(E_\nu) \int_{z_p^{\min}}^{z_p^{\max}} dz_p \int_{z_e^-}^{z_e^+} dz_e \frac{4E_\nu \mathcal{M}_{\phi_0}}{\sqrt{S^2 - (E_\nu^2 - R)^2}} \,. \tag{43}$$

The second line corresponds to the *I* integral (20), (42). It is natural to group a factor of $E_\nu$ with $\mathcal{M}_{\phi_0}$ to cancel the factor of $E_\nu$ in the denominator (30) which can cause numerical problems at the edge of the kinematically allowed momentum range where $E_\nu \to 0$.

The neutron decay rate can therefore be computed as a 5-dimensional momentum integral (43), obtaining the integration ranges from (21), (22), (38) and (41), the matrix element from (30), and the Jacobian (square root denominator) from (24), (25).

## C. Modified Urca Neutron Decay Rate

The matrix element is (4.16) in [31,60]

$$\left(s\frac{\sum|M_n|^2}{2^6 E_n^* E_p^* E_e E_\nu E_{N_1}^* E_{N_2}^*}\right) = 42G^2 \frac{f^4}{m_\pi^4}\frac{g_A^2}{E_e^2}\frac{k_{Fn}^4}{(k_{Fn}^2 + m_\pi^2)^2}, \tag{44}$$

where $f \approx 1$ is the N-$\pi$ coupling and $s = 1/2$ for the identical particles. The conventional way of doing the integral is to divide the integral into an energy integral and an angular integral (termed "phase space decomposition" [35])

$$\int dk_n^3 dk_p^3 dk_e^3 dk_\nu^3 dk_{N_1}^3 dk_{N_2}^3 = \int dk_n dk_p dk_e dk_\nu dk_{N_1} dk_{N_2} k_n^2 k_p^2 k_e^2 k_\nu^2 k_{N_1}^2 k_{N_2}^2$$

$$\times \int d\Omega_n d\Omega_p d\Omega_e d\Omega_\nu d\Omega_{N_1} d\Omega_{N_2}. \tag{45}$$

We use relativistic dispersion relations for nucleons

$$E_N = \sqrt{k^2 + m_N^{*2}} + U_N, \tag{46}$$

where U is the mean field contribution to the energy. We define $E^* \equiv \sqrt{k^2 + m^{*2}}$, then $dE^* = kdk/E^*$. We use ultrarelativistic dispersion relations for electron and neutrino,

$$E = k, \tag{47}$$

then $dE = dk$ (the electron mass $m_e = 0.511\,\text{MeV}$ is negligible compared to its momentum). Therefore, we can convert the momentum integral to an energy integral, and the rate integral becomes

$$\Gamma_{mU,nd(n)} = \frac{42G^2 g_A^2 f^4}{(2\pi)^{14} m_\pi^4} \int d\Omega_n d\Omega_p d\Omega_e d\Omega_\nu d\Omega_{N_1} d\Omega_{N_2}$$

$$\times \delta^{(3)}(\vec{k}_n + \vec{k}_{N_1} - \vec{k}_p - \vec{k}_e - \vec{k}_{N_2})k_n^2 k_p^2 k_e^2 k_\nu^2 k_{N_1}^2 k_{N_2}^2 \frac{1}{E_e^2}\frac{k_{Fn}^4}{(k_{Fn}^2 + m_\pi^2)^2}$$

$$\times \int dE_n^* dE_p^* dE_e dE_\nu dE_{N_1}^* dE_{N_2}^* \frac{E_n^*}{k_n}\frac{E_p^*}{k_p}\frac{E_{N_1}^*}{k_{N_1}}\frac{E_{N_2}^*}{k_{N_2}}$$

$$\times \delta(E_n + E_{N_1} - E_p - E_e - E_\nu - E_{N_2})f_n f_{N_1}(1 - f_p)(1 - f_e)(1 - f_{N_2}). \tag{48}$$

Notice that it is most common to set $\vec{k}_\nu = 0$ in the momentum conserving delta function but keep $E_\nu$ in the energy delta function.

In the Fermi surface approximation, we set all momenta to Fermi momenta and we will have $E_e = k_e = k_{Fe}, k_\nu = E_\nu$.

Now, the rate integral becomes

$$\Gamma_{mU,nd(n)} = \frac{42G^2 g_A^2 f^4}{(2\pi)^{14} m_\pi^4}k_{Fn}^2 k_{Fp}^2 k_{Fe}^2 k_{FN_1}^2 k_{FN_2}^2 \frac{1}{k_{Fe}^2}\frac{k_{Fn}^4}{(k_{Fn}^2 + m_\pi^2)^2}\frac{E_n^*}{k_{Fn}}\frac{E_p^*}{k_{Fp}}\frac{E_{FN_1}^*}{k_{FN_2}}\frac{E_{N_2}^*}{k_{N_2}}$$

$$\times \int d\Omega_n d\Omega_p d\Omega_e d\Omega_\nu d\Omega_{N_1} d\Omega_{N_2}\delta^{(3)}(\vec{k}_n + \vec{k}_{N_1} - \vec{k}_p - \vec{k}_e - \vec{k}_{N_2})$$

$$\times \int dE_n^* dE_p^* dE_e dE_\nu dE_{N_1}^* dE_{N_2}^* E_\nu^2 f_n f_{N_1}(1 - f_p)(1 - f_e)(1 - f_{N_2})$$

$$\times \delta(E_n^* + E_{N_1}^* - E_p^* - E_e - E_\nu - E_{N_2}^* + (U_n - U_p)). \tag{49}$$

For the energy integral, we do a change of variable,

$$x = \frac{E^* - \mu^*}{T},$$  (50)

then, $dx = (1/T)dE^*$ and $\mu = 0$ for the neutrino. For the integral bounds, we have

$$\int_{m^*}^{+\infty} dE^* = T \int_{(m^* - \mu^*)/T}^{+\infty} dx = T \int_{-(\mu^* - m^*)/T}^{+\infty} dx \approx T \int_{-\infty}^{+\infty} dx,$$  (51)

where the approximation is valid because $\mu^* \gg T$. For neutrino, $\mu = 0$ and $m = 0$, so the lower bound is 0. Then, the energy integral, which we denote as $I$, becomes

$$\begin{aligned}
I \equiv & \int dE_n^* dE_p^* dE_e dE_\nu dE_{N_1}^* dE_{N_2}^* E_\nu^2 f_n f_{N_1} (1 - f_p)(1 - f_e)(1 - f_{N_2}) \\
& \times \delta(E_n^* + E_{N_1}^* - E_p^* - E_e - E_\nu - E_{N_2}^* + (U_n - U_p)) \\
= & T^7 \int dx_n dx_p dx_e dx_\nu dx_{N_1} dx_{N_2}\, x_\nu^2 f(x_n) f(x_{N_1})(1 - f(x_p))(1 - f(x_e)) \\
& \times (1 - f(x_{N_2})) \delta(x_n + x_{N_1} - x_p - x_e - x_\nu - x_{N_2} + \frac{\mu_n - \mu_p - \mu_e}{T}) \\
= & T^7 \int_0^{+\infty} dx_\nu x_\nu^2 \int_{-\infty}^{+\infty} dx_n dx_p dx_e dx_{N_1} dx_{N_2}\, f(x_n) f(x_{N_1}) f(-x_p) f(-x_e) \\
& \times f(-x_{N_2}) \delta(x_n + x_{N_1} - x_p - x_e - x_\nu - x_{N_2} + \frac{\mu_n - \mu_p - \mu_e}{T}) \\
= & T^7 \int_0^{+\infty} dx_\nu x_\nu^2 \int_{-\infty}^{+\infty} dx_n dx_p dx_e dx_{N_1} dx_{N_2}\, f(x_n) f(x_{N_1}) f(x_p) f(x_e) f(x_{N_2}) \\
& \times \delta(x_n + x_{N_1} + x_p + x_e - x_\nu + x_{N_2} + \frac{\mu_n - \mu_p - \mu_e}{T}).
\end{aligned}$$  (52)

One can use Mathematica to obtain an analytical expression,

$$I = \frac{1}{12} F(\xi),$$  (53)

where $\xi \equiv (\mu_n - \mu_p - \mu_e)/T$, and

$$\begin{aligned}
F(\xi) \equiv & - (\xi^4 + 10\pi^2\xi^2 + 9\pi^4)\mathrm{Li}_3(-e^\xi) + 12(\xi^3 + 5\pi^2\xi)\mathrm{Li}_4(-e^\xi) \\
& - 24(3\xi^2 + 5\pi^2)\mathrm{Li}_5(-e^\xi) + 240\xi\mathrm{Li}_6(-e^\xi) - 360\mathrm{Li}_7(-e^\xi).
\end{aligned}$$  (54)

For the angular integral, we can look up [25], which calculated the n-dimensional angular integral for n=3,4,5, and obtain

$$A = \frac{32\pi(2\pi)^4}{k_n^3} \theta_n,$$  (55)

where

$$\theta_n = \begin{cases} 1 & k_{Fn} > k_{Fp} + k_{Fe} \\ 1 - \dfrac{3}{8} \dfrac{(k_{Fp} + k_{Fe} - k_{Fn})^2}{k_{Fp} k_{Fe}} & k_{Fn} < k_{Fp} + k_{Fe}. \end{cases}$$  (56)

Therefore, the neutron decay modified Urca rate with n-spectator under Fermi surface approximation is

$$\Gamma_{mU,nd(n)}(\xi) = \frac{7}{64\pi^9} G^2 g_A^2 f^4 \frac{(E_{Fn}^*)^3 E_{Fp}^*}{m_\pi^4} \frac{k_{Fn}^4 k_{Fp}}{(k_{Fn}^2 + m_\pi^2)^2} F(\xi) T^7 \theta_n.$$  (57)

Similarly, we can calculate the electron capture mU rate with n-spectator

$$\Gamma_{mU,ec(n)}(\xi) = \Gamma_{mU,nd(n)}(-\xi) . \tag{58}$$

For p-spectator processes, the matrix element is

$$\left( s \frac{\sum |M_p|^2}{2^6 E_n^* E_p^* E_e E_\nu E_{N_1}^* E_{N_2}^*} \right) = 48 G^2 \frac{f^4}{m_\pi^4} \frac{g_A^2}{E_e^2} \frac{(k_{Fn} - k_{Fp})^4}{((k_{Fn} - k_{Fp})^2 + m_\pi^2)^2} , \tag{59}$$

where we still have $s = 1/2$. Then we have the mU rates with p-spectator

$$\Gamma_{mU,nd(p)}(\xi) = \frac{1}{64\pi^9} G^2 g_A^2 f^4 \frac{(E_{Fp}^*)^3 E_{Fn}^*}{m_\pi^4} \frac{(k_{Fn} - k_{Fp})^4 k_{Fn}}{((k_{Fn} - k_{Fp})^2 + m_\pi^2)^2} F(\xi) T^7 \theta_p , \tag{60}$$

$$\Gamma_{mU,ec(p)}(\xi) = \Gamma_{mU,nd(p)}(-\xi) , \tag{61}$$

where

$$\theta_p = \begin{cases} 0 & k_{Fn} > 3k_{Fp} + k_{Fe} \\ \dfrac{(3k_{Fp} + k_{Fe} - k_{Fn})^2}{k_{Fn}k_{Fe}} & 3k_{Fp} + k_{Fe} > k_{Fn} > 3k_{Fp} - k_{Fe} \\ \dfrac{4(3k_{Fp} - k_{Fn})}{k_{Fn}} & 3k_{Fp} - k_{Fe} > k_{Fn} > k_{Fp} + k_{Fe} \\ 2 + \dfrac{3(2k_{Fp} - k_{Fn})}{k_{Fe}} - \dfrac{3(k_{Fn} - k_{Fe})^2}{k_{Fn}k_{Fe}} & k_{Fn} < k_{Fp} + k_{Fe} . \end{cases} \tag{62}$$

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
