# Peer review of "Beta Equilibrium under Neutron Star Merger Conditions"

_universe, doi:10.3390/universe7110399_

Round 1

Reviewer 1 Report

The authors of "Beta equilibrium under neutron star merger conditions" consider relativistic mean field models of nuclear matter at non-zero temperature and discuss the conditions needed to maintain beta equilibrium. The authors improve previous calculations of the isospin chemical potential required to achieve beta equilibrium for select nuclear matter models. The work is good solid work and likely of interest to the audience of Universe.

While I do not have objection of the technical results obtained in this paper, I find that there are some general issues that need to be addressed.

1) The authors consider (small) temperatures, T\sim 1-5 MeV, because then the direct Urca threshold is apparent. However, from the text one gets an impression that these are maximum temperatures reached in neutron star mergers. As shown by several groups, the temperatures are not only moderately larger >10MeV where the results of this paper are not valid anymore but can even reach tens of MeV's. The authors can consult: Oeschlin et al . astro-ph/0611047 ;  Baioitti-Rezzolla 1607.03540 ;  Most et al. 1807.03684 ;  Raithel et al. 2104.07226 . In addition, in ref. [1] the authors' list a statement appears "...and temperatures up to T ∼ 50 − 100 MeV in the remnant", with other relevant literature cited therein. The statements e.g. in the abstract, introduction, and section 2 need to reflect this more clearly. This also puts the choice of the title under question. One should at least add comments already in the introduction that this work does not establish that beta equilibrium is maintained in neutron star merger conditions. 

2) I notice that IUF is in mild tension with 2 solar mass limit, i.e. at 1\sigma as probably noticed by the authors in sec. 3. Is there a particular reason that the authors did not consider, e.g. DD2, another finite-T relativistic mean field model available in CompOSE and that was considered in [5] in the previous approximation by two of the authors of the present work? DD2 is similarly only mildly in tension with astrophysical observations (tidal deformability constraint by Ligo/VIRGO), so I do not see a reason why it is disregarded. 

Reviewer 2 Report

Main remark:

It seems to me that the authors need to add to the text an explanation concerning whether the isospin  chemical potential $\mu_{\delta}$ is an empirical fitting parameter leading to equilibrium between the two direct Urca processes or this quantity can be determined independently (thermodynamically). In the second case , this value should follow from the expression , e.g. for the differential  of  thermodynamic energy.

Small note:

Line 216  --  the authors need to replace the  word "capture" (neutron capture)by the word "decay'  (neurton decay).

Suggestion:

It would be easier for the reader if Fig. 8 were moved to some place before Fig. 9, but not after Fig.12, as it is now.

Reviewer 3 Report

The paper presents a calculation of the isospin chemical potential needed for reaching beta-equilibrium in neutron star matter in a temperature window, 1-5 MeV, for which neutrinos are not trapped and the standard beta-equilibrium conditions fail. The corresponding Urca rates are modified by an order of magnitude or more with respect to the case in which the isospin imbalance is neglected. Two models for the equation of state are used with the corresponding dispersion relations.

The paper is interesting, rather technical but very useful for people performing numerical simulations of the merger of compact stars and trying to treat in a “simple” scheme neutrino transport. Indeed, a full neutrino transport treatment would be preferable but at the moment such a complicated simulation has not yet been performed. The paper can be published in the present form, I would just ask to authors to discuss how their results could be used in numerical schemes and compare for instance with presently used approximations such as the neutrino leakage scheme, see e.g. Astrophys.J. 869 (2018) 2, 130.
